# Adaptive Exoskeleton Device for Stress Reduction in the Ankle Joint Orthosis

**DOI:** 10.3390/s25030832

**Published:** 2025-01-30

**Authors:** Andrey Iziumov, Talib Sabah Hussein, Evgeny Kosenko, Anton Nazarov

**Affiliations:** Department of the Robotics and Mechatronics, Faculty of Automation, Mechatronics and Control, Don State Technical University, 344003 Rostov-on-Don, Russia; talibsabah1981@gmail.com (T.S.H.); a123lok@mail.ru (E.K.); mrkombat011@gmail.com (A.N.)

**Keywords:** robotic sensor systems, sensor arrays, force sensors, tactile sensors, foot pressure, interventional pressure

## Abstract

Treating ankle fractures in athletes, commonly resulting from training injuries, remains a significant challenge. Current approaches to managing both non-surgical and postoperative foot and ankle disorders have focused on integrating sensory systems into orthotic devices. Recent analyses have identified several gaps in rehabilitation strategies, especially regarding gait pattern reformation during recovery. This work aims to enhance rehabilitation effectiveness for patients with ankle injuries by controlling load distribution and monitoring joint flexion/extension angles, as well as the reactive forces during therapeutic exercises and walking. We developed an exoskeleton device model using SolidWorks 2024 software, based on data from two patients: one healthy and one with an ankle fracture. Pressure measurements in the posterior limb region were taken using the F-Socket system and a custom electromechanical sensor designed by the authors. The collected data were analyzed using the butterfly parameterization method. This research led to the development of an adaptive exoskeleton device that provided pressure distribution data, gait cycle graphs, and a diagram correlating foot angles with the duration of exoskeleton use. The device demonstrated improvement in the patients’ conditions, facilitating a more normalized gait pattern. A reduction in the load applied to the ankle joint was also observed, with the butterfly parameter confirming the device’s correct operation.

## 1. Introduction

Ankle fractures commonly result from sports-related injuries, ligament sprains, or accidental falls. The treatment approach depends on the type of fracture and specific patient factors. Even when surgically treated, healing may take considerable time, influenced by various factors such as the fracture type, the patient’s age, comorbidities, and rehabilitation protocols.

Immobilization plays a crucial role in ankle fracture rehabilitation, aiming to reduce stress on the joint by applying a splint or plaster cast for a specified period. However, traditional methods often lead to complications, the most common being joint stiffness.

To address this issue, the present work introduces a prototype exoskeleton device designed by combining two types of ankle orthoses. This device supports joint protection while enhancing rehabilitation efficacy, specifically reducing stiffness risk. It allows controlled movement of the ankle joint (in both dorsiflexion and plantar flexion) in the sagittal plane, manages foot alignment angles, reduces load on the ankle joint, alleviates pain, and protects against adverse side effects, significantly supporting the rehabilitation process.

Orthotics play a pivotal role in the treatment of both non-surgical and post-surgical foot and ankle conditions. Increasingly, ‘off-the-shelf’ customized orthotics are replacing traditional plaster fixation in the treatment of musculoskeletal issues of the ankle and hindfoot. These orthotics offer various levels of support, enabling patients to remain mobile during rehabilitation.

Many studies by international researchers have contributed to this field. One prominent approach involves integrating sensor systems into orthoses, including force-sensitive sensors. In classical robotics, sensor integration is widely used in the development of robotic arms, gloves, and flexible robots for human–machine interaction [1,2]. Such advancements are also applied in industries like automotive engineering, where flexible sensors monitor tire surface pressure and contact with the road [3].

Force-sensitive sensors are extensively utilized in EMS applications, including devices for gait analysis, anomaly detection, and monitoring of conditions like diabetic foot [4,5]. These sensors are also crucial in rehabilitation measurement systems, particularly for strain-tremor assessments [6], and are fundamental in developing exoskeletons for elderly care, focusing on system control and patient interaction [7]. Research has also explored sensor materials such as carbon nanotubes and polydimethylsiloxane to create force-sensitive sensors with promising results in electromagnetic protection and high conductivity [8,9].

The field of rehabilitation robotics heavily relies on regular data collection and analysis, including the monitoring of physical activity and sports performance. For example, integrating sensors into sports equipment to enhance performance and analyze movements, as well as to evaluate the effectiveness of orthoses, has shown significant promise in sports EMS applications [10,11].

In the context of monitoring and control systems, human–machine interfaces for analyzing and processing sensor data are gaining popularity. Machine learning algorithms, for instance, have significantly improved the accuracy of sensor data classification and analysis in both EMS and sports technologies [12,13]. Additionally, research [14] has demonstrated methods to mitigate the impact of misalignment in human–machine interfaces, improving user comfort and overall device functionality.

Advancements in the development of ultra-sensitive pressure sensors, as described in [15], have opened new possibilities for precise load monitoring in rehabilitation systems. Simultaneously, the use of intelligent technologies in the fabrication of orthopedic insoles [16] has significantly enhanced the adaptability and functionality of such devices, improving their interaction with the operator.

The advancement of these scientific approaches has influenced the development and testing of new platforms. For instance, the introduction of innovative techniques in the design of high-performance sensors has enhanced their stability and sensitivity. The application of these concepts in exoskeletal devices for lower limbs has experimentally demonstrated improvements in rehabilitation processes, particularly for older adults, aligning with the WHO’s healthy aging initiative [17]. Comprehensive technical solutions for restoring lower limb function are employed in patients with hemiplegia, where the precise tracking of predetermined movement paths is essential [18].

This review highlights gaps in current research on integrating sensory systems with orthoses, underscoring the need for further advancements in this area. While innovations in sensory technologies and their application in EMS and engineering fields have the potential to significantly improve the quality of life for users and the effectiveness of therapeutic procedures, certain challenges remain unresolved. For instance, many researchers have not sufficiently examined how different robotic support devices impact patients’ subjective complaints or how various exoskeletal models influence gait formation during rehabilitation. Additionally, support braces do not adequately reduce the stress on the foot and ankle caused by external forces. Several questions still need to be addressed, which may limit the practical application of these devices during the healing period or rehabilitation procedures—questions that this thesis seeks to explore.

To achieve the research objective, several specific aims have been identified:
Identification of the cause-and-effect relationships between joint movement, human muscular activity, and the motion of exoskeletal system components.Analysis of the impact of individual operator characteristics on the control and performance of the exoskeleton device.Enhancement of fatigue resistance and structural durability in the proposed exoskeleton model.Comparative assessment of the patient’s condition before and after using the proposed exoskeleton model, to determine the most suitable design, based on the following:
a: an experimental test;b: the opinion of the patient’s attending physician.

The novelty of the results presented in this manuscript is demonstrated through the following contributions:
Development of a cost-effective and highly accurate force sensor: This study introduces a novel force sensor that is significantly more affordable than existing models, such as the socket-type F sensor, while offering comparable or superior accuracy.Creation of an active orthosis with simplified mechanics: The orthosis features an intuitive and streamlined design, enhancing ease of use for patients. An adjustment lever, mounted on the ankle joint, enables users to independently customize the device to their specific needs without requiring complex configurations.Compliance with biomechanical requirements: Treadmill tests have shown that the orthosis facilitates movement closely resembling that of a healthy foot. This is supported by the results of the “butterfly” test, which compares the gait patterns of the patient wearing the exoskeleton to those of a healthy individual.

## 2. Materials and Methods

This study focuses on the design of orthopedic models for the lower limbs, intelligent motion systems, statistical data analysis, and the modeling of kinematic and dynamic characteristics.

Data were collected through tests on two subjects with identical physical characteristics (height and weight): one being a healthy individual, and the other suffering from an ankle fracture caused by a sports injury. The exoskeleton device model was developed using SolidWorks software, while a finite element model (FEM) of the device was created and analyzed using ANSYS 2021 R2 software.

This research is distinguished by its international collaboration. At Al-Nahrain University, the Department of Prosthetics and Orthotics, in conjunction with the Mechatronics Laboratory, tests were conducted to measure the interface pressure (IP) and force between the foot and the orthosis using a custom-built electromechanical sensor (EMS) [19]. The reliability of the results is ensured by the use of modern sensors, including the F-Socket system and finite element analysis, with theoretical conclusions aligning with real-world experimental outcomes. Additionally, the performance of the custom EMS was compared against that of the F-Socket sensor, with findings corroborated by studies presented in [20,21]. The key results were presented and discussed at several conferences, notably the 4th International Scientific Conference of Engineering Sciences and Advanced Technologies (IICESAT) in Iraq (Babylon University, 2022) [22].

It is important to note that the exoskeleton described in this study was custom designed according to the individual parameters of a patient with traumatic injuries, specifically an ankle fracture (Figure 1).

The normal position of the human ankle on the foot is 90°. From the ankle position, the range of dorsiflexion is 0°–20° and plantarflexion is 0°–45° [23]. The value of the AFO angle can be controlled manually by tightening or loosening the regulator nut, which connects the electromechanical sensor arm and the hinge joint fixed on the support platform. To ensure sufficient motion during walking, the AFO was mounted with a support platform at an angle of 25°. This position allows the movement to be positioned in the sagittal plane and dissipates the force applied to the ankle joint, pushing it towards the forefoot.

The exoskeleton should simulate the movement of the human ankle joint, by moving the (AFO) in the range from 0° dorsiflexion to 20° plantarflexion in the sagittal plane, as shown in Figure 2. This is used to minimize the ankle stiffness problem, speed up the recovery process, and protect it from side effects.

The ankle exoskeleton utilizes materials such as perlon, carbon fiber, and polypropylene, selected for durability, resistance, and cost-effectiveness. In this study, polypropylene was chosen due to the following characteristics:
Semi-rigid material and high temperature resistance;Transparent material and high chemical resistance;Durable material and high wear resistance;Low cost.

The exoskeleton’s computer model was created using SolidWorks software, which enables seamless integration with ANSYS Workbench 2021.R2 through file formats such as SLDPRT, SLDASM, ACIS (.sat), and Mechanical Desktop (.dwg). This facilitates efficient workflows between CAD and simulation platforms.

To enhance mobility and ease of use, the device is designed to be user-friendly and ergonomic, allowing patients to easily don and remove the orthosis while securing it properly to the lower limb.

Stainless steel was selected for manufacturing the electromechanical sensor (Figure 3a,c), while the Inconel 718 alloy was chosen for the support platform model of the exoskeleton device (Figure 3b).

The EMS works as follows: The action of the force sensor is activated when the force sensor is subjected to any movement by contacting a solid ball (9) in the head of the piston arm (6) with any external cross-section, and the base of the piston arm (7) moves the handle (4) of the resistor (3) with stops (5), with the handle (4) attached to the slot (8) of the base of the piston arm (7) trapped between the first spring (10) and the second spring (11). On the other hand, by moving backward to change the value of the variable resistor (10 KΏ, 3) according to the value of the internal displacement, the base of the piston arm (7) is raised and returns to its first position by the action of a second spring (11), which simultaneously pushes the base of the piston arm (7) and the handle (4) of the resistor (3). Figure 3a,c show a model drawing of the EMS manufactured.

When selecting actuators, several key factors must be considered, including size, efficiency, load capacity, weight, and cost. In the proposed device, the actuator is a spring-based mechanism. The selection of the springs is directly related to the calculation of the load applied during ankle joint movement, specifically during flexion and extension. The load is determined by the following equation:
F = k(*L*_0_ − *L*_1_),(1)
where F is the load in Newtons (N), *L*_0_ is the unloaded spring length in m, *L*_1_ is the loaded spring length in m, and k is the stiffness coefficient in N/m. In the previous expression, the stiffness coefficient k is determined by the following formula:k = Gd^4^/8nD^3^,(2)
where k is the stiffness coefficient of spring; G is the modulus of elasticity of the spring material in MPa; d is the diameter of the wire in mm; n is the number of coils of the spring; D is the average diameter of the spring in m; d is the diameter of the wire in mm; and n is the number of coils of the spring.

Figure 4 shows the adaptive load control algorithm with the pressure calculation.

Below, we describe how the algorithm works:
Start.Device Initialization and Sensor Check:
If sensors are functional, proceed to the next step;If sensor errors are detected, trigger an error alert and terminate the operation.Data Collection on Current State (pressure, flexion angle):
Calculate force F;If pressure and force are within the acceptable range, proceed to step 5;If pressure exceeds the range or the angle is out of range, proceed to step 4.Adjust Exoskeleton Parameters:
Adjust the angle or pressure to ensure load values are within acceptable limits;Return to step 3 to remeasure.Evaluate Patient Condition:
If the patient reports discomfort, proceed to step 4 for adjustment;If the patient is comfortable, proceed to step 6.Save Data and Complete Cycle.End Cycle.

The electromechanical device (EMS) contains two springs: a 0.095 m compression spring located in front of the piston and a 0.065 m extension spring placed behind the piston, which returns the device to its neutral position during unloading. Both springs have a diameter of 16 mm. The compression spring has an elastic force of 47.98 N, while the extension spring exerts 70.21 N.

The device, an electromechanical actuator, is capable of withstanding high-pressure loads. The actuator spring must be robust enough to maintain the stem position and withstand the forces generated by the human body. An electromechanical actuator is preferred for this application due to its self-operating capabilities, which differentiates it from pneumatic actuators. A normal person can exert forces up to 1.5 times their body weight when walking. This can be up to 800–1200 N with an average person weighing approximately 80–120 kg [24].

A preliminary calculation was conducted before patient testing, establishing that the maximum force exerted by the actuator is approximately 1200 N, corresponding to a load of 120 kg. These calculated values, based on the mechanical properties of the materials, were used in the design and fabrication of the support platform, and electromechanical sensor.

Standard grade 8.8 bolts and nuts were employed to assemble the device. The materials used in the exoskeleton include standard polypropylene, stainless steel, Inconel 718, and aluminum 2024T351. The components of the exoskeleton are shown in Figure 5.

## 3. Experimental Work

The kinematic and kinetic data during the experimental work were obtained from two young subjects; one of them had an injury and the second was healthy. The first was approximately 26 years old, 168 cm in height, and weighed 60 kg, and the second was approximately 24 years old, 170 cm in height, and weighed 60 kg. Both subjects wore the exoskeleton device on the lower right limb, as shown in Figure 6.

The pressure measurements were obtained by using two experimental methods. The first method employed the F-Socket sensor and the second employed the EMS; both methods were employed to measure the pressure and force between the foot and the exoskeleton.

Measurements were taken in the posterior region of the limb using both methods, as illustrated in Figure 7. The results from the two methods were compared to validate the performance of the fabricated system.

In addition, the gate cycle results (GRF, pressure distribution, center of pressure (COP), candice, step length, and foot-print analysis) and safety factor, von Mises stress analysis, deformation, and service life of the exoskeleton device model are illustrated in this study, as shown in Table 1, Table 2, Table 3, Table 4 and Table 5 and Figure 8, Figure 9, Figure 10, Figure 11, Figure 12, Figure 13, Figure 14, Figure 15, Figure 16, Figure 17, Figure 18, Figure 19, Figure 20, Figure 21, Figure 22 and Figure 23.

## 4. Results and Discussion

The studies conducted on the application of the exoskeleton device showed the following results:

F-Socket Sensor Measurement Method. The measurement results indicated that the maximum interventional pressure (IP) between the exoskeleton and the foot reached 156 kPa, while the corresponding force was 1335 N. A summary of the experimental data is provided in Figure 8 and Table 1.

Manufactured EMS sensor measurement method. The measurement results indicated that the maximum interventional pressure (IP) between the exoskeleton and the foot reached 152 kPa, while the corresponding force was 1328 N. A summary of the experimental data is provided in Figure 9 and Table 2.

Figure 10 illustrates the stance phase event. The GRF curve can be divided into three sections or parts. The first stage starts with the first peak and represents the initial contact, the second stage represents the mid-stance where the body load is distributed over the entire foot area, and the third stage starts with the first or second peak and represents the toe-off position. All of these sections or parts of the GRF were created as a result of stance phase events. The big difference between the right and left foot of the patient can be noticed as a result of the patient relying on the healthy foot (left foot) while walking, as shown in Figure 12 and Figure 13, compared with the results of the curves obtained for the healthy person, which are almost identical due to the normal distribution of body weight on both feet (left and right foot), as shown in Figure 10.

In a healthy subject, the GRF curve exhibits a characteristic dip, as shown in Figure 11, caused by the normal distribution of body weight between the heel and forefoot. In contrast, the pressure gradually accumulates in the affected foot region.

In Figure 12, we notice the asymmetry of the curve pattern for the left and right foot due to the reduced contact area between the foot and the floor, while we notice the similarity of the curve pattern for the left and right foot in the case of the patient wearing the exoskeleton due to the increased contact surface area between the foot and the floor, as shown in Figure 13.

Figure 14, Figure 15 and Figure 16 show the pressure distribution patterns for both healthy and pathological conditions in the left and right lower limbs. Figure 15 illustrates a higher level of pressure on the left foot compared to the right. This discrepancy is attributed to the reduced contact area between the patient’s right shoe and the treadmill surface, as the patient predominantly used the forefoot due to ankle pain. In Figure 15, we notice the asymmetry of the curve pattern for the left and right foot due to the reduced contact area between the foot and the floor, while we notice the similarity of the curve pattern for the left and right foot in the case of the patient wearing the exoskeleton due to the increased contact surface area between the foot and the floor, as shown in Figure 16.

The butterfly shape refers to the abduction and adduction of the foot during walking, the direction of the center of pressure (COP) for both cases (healthy and pathological) for the left and right feet, and these graphs depict the COP during movement, including the behavior of the foot from heel contact to the toe-off position. As a result, MPV occurs at each of these stages. There is congruence between the left and right feet in the graph of the gait pattern of the healthy subject, as shown in Table 3 and Figure 17, although there is some difference between the gait pattern of the left and right feet. In the case of the patient, we see a difference between the gait of the left and right foot when walking with the exoskeleton and without the exoskeleton device, as shown in Table 4 and Figure 18. When the patient wears the exoskeleton, we notice an improvement in the patient’s gait pattern or the resemblance of butterfly wings. This shows the effectiveness of the manufactured exoskeleton, as shown in Table 5 and Figure 19.

Results of numerical analysis. Finite element method (FEM) software was used to determine the equivalent safety stress and von Mises stress for the analysis of the exoskeleton model. The yield strength, according to von Mises’ theory, is determined by the following criteria: (σe < σy, safe), (σe = σy, critical), and (σe > σy, failure), where σe = stress equivalent and σy = yield strength. The strength safety calculation must be equal to or greater than 1.25.

Exoskeleton model at maximum load. Figure 20, Figure 21, Figure 22 and Figure 23 show the von Mises stress, deformation, safety factor, and service life the exoskeleton model, respectively. The exoskeleton model was subjected to a maximum load of 1200 N.

## 5. Conclusions

This study successfully designed and fabricated a prototype exoskeleton device that integrates the kinetic properties of both fixed and mobile orthoses. The device facilitated an in-depth investigation of the interaction between the exoskeleton and its operator, including tests with healthy individuals and patients with ankle injuries.The exoskeleton significantly enhanced gait parameters in patients. Experimental results demonstrated that the patients’ gait patterns approached normal levels when utilizing the exoskeleton. Furthermore, performance comparisons revealed that the custom-built electromechanical sensor (EMS) achieved results comparable to the commercially available F-Socket sensor, with maximum pressure and force measurements of 156 kPa and 1335 N for the F-Socket sensor, and 152 kPa and 1328 N for the EMS sensor. The EMS sensor exhibited sufficient accuracy for rehabilitation purposes.This work established causal relationships between the motor activity of the ankle joint and the movement of exoskeleton components, highlighting the mechanical synergy between the patient’s movements and the device’s functionality.The primary objectives of this study—reducing ankle joint loads, accelerating recovery phases, and ensuring joint protection—were effectively achieved. The analysis based on butterfly-type graph metrics demonstrated notable improvements in gait parameters for patients with pathological conditions, affirming the device’s operational effectiveness.The exoskeleton’s performance was verified through comprehensive mechanical testing, including von Mises stress analysis, strain evaluation, fatigue safety margin assessment, and projected service life estimation. The calculated durability suggests a lifespan of approximately 40,447 gait cycles, or 5.5 years, assuming three rehabilitation sessions per week with 150 steps per session. This underscores the exoskeleton’s long-term utility and robustness in rehabilitation therapy.

## Figures and Tables

**Figure 1 sensors-25-00832-f001:**
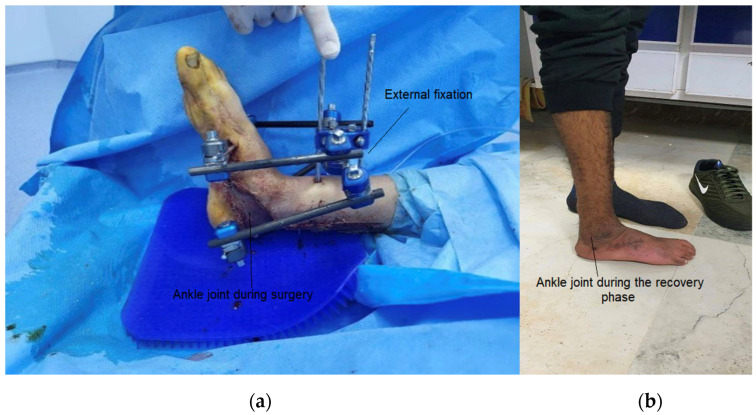
Condition of the patient’s foot (**a**) before the use of the device and (**b**) after the rehabilitation process.

**Figure 2 sensors-25-00832-f002:**
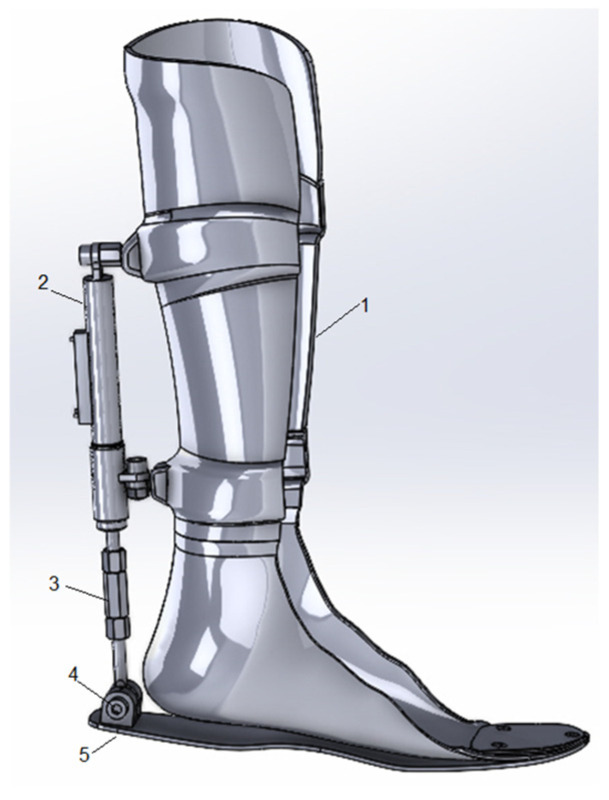
Design of the exoskeleton model by using SolidWorks software. (1) Orthosis, (2) electromechanical sensor, (3) height regulator, (4) hinge joint, and (5) support platform.

**Figure 3 sensors-25-00832-f003:**
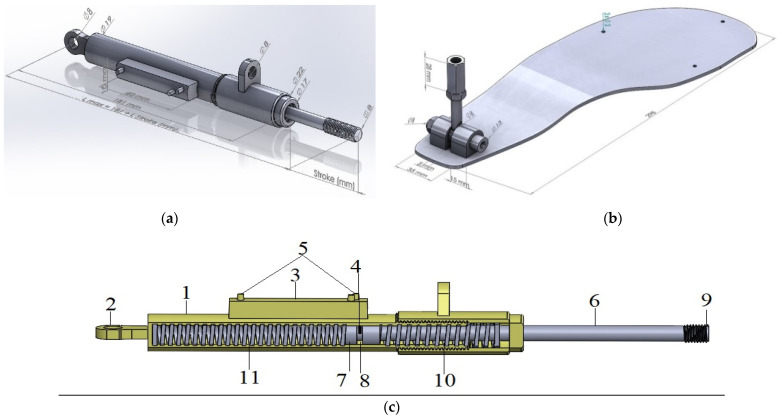
Electromechanical sensor (**a**,**c**) and (**b**) support platform with the hinge joint.

**Figure 4 sensors-25-00832-f004:**
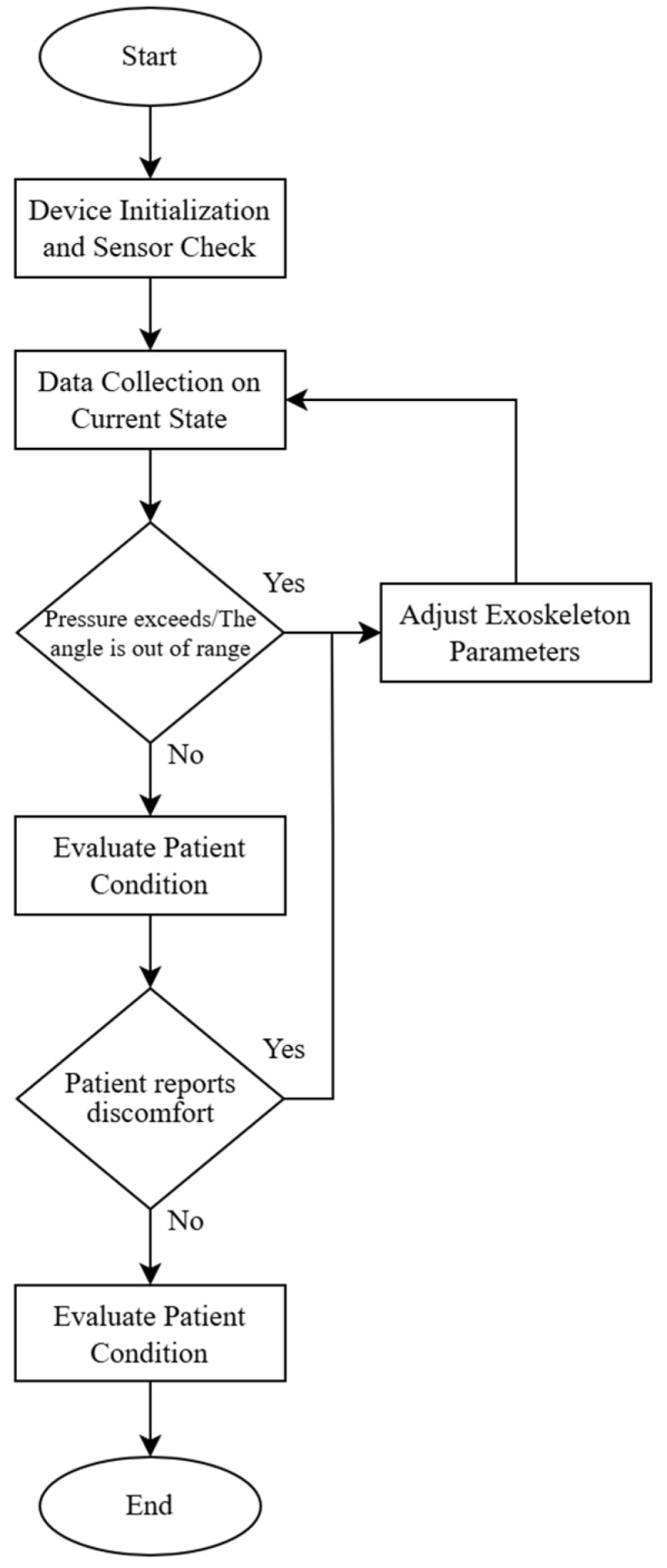
Adaptive load control algorithm with pressure calculation.

**Figure 5 sensors-25-00832-f005:**
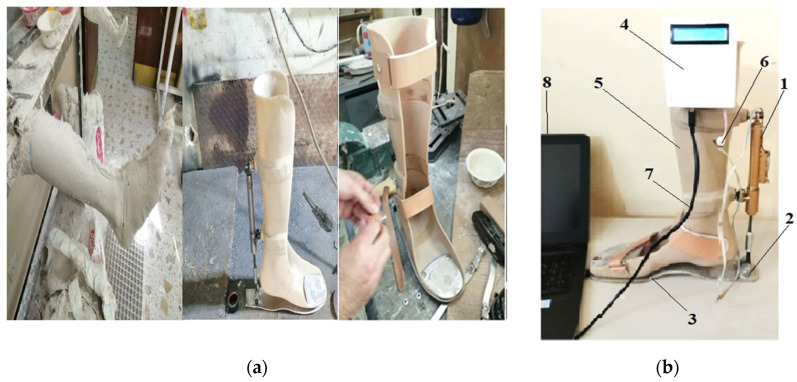
(**a**) Fabrication stages and (**b**) appearance of the exoskeleton device at the final stage, where 1—EMS, 2—hinge joint, 3—support platform, 4—microcontroller unit, 5—orthosis (AFO), 6—wire to microcontroller unit, 7—wire to computer, and 8—computer.

**Figure 6 sensors-25-00832-f006:**
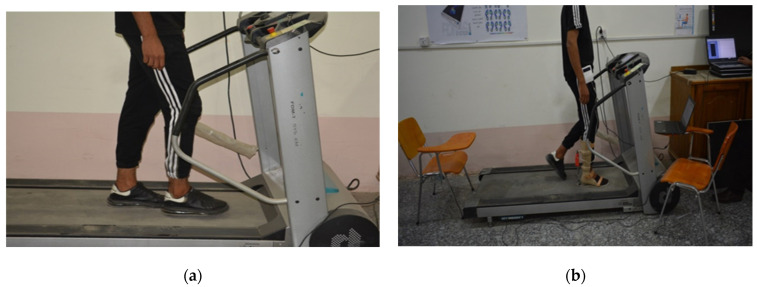
(**a**) Patient subject walking on the treadmill device without wearing the exoskeleton and (**b**) with the manufactured exoskeleton device.

**Figure 7 sensors-25-00832-f007:**
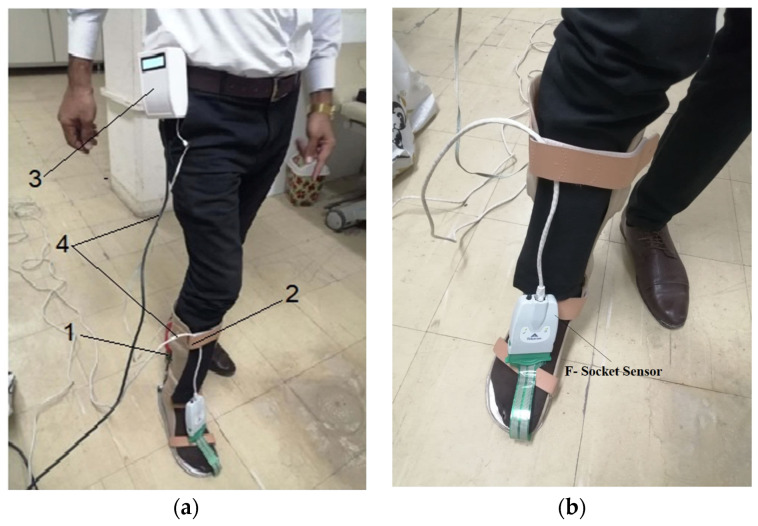
Experimental interventional pressure (IP) and force measurement: (**a**) EMS measurement method, where 1—electromechanical sensor, 2—exoskeleton, 3—microcontroller unit, and 4—wires to computer; (**b**) F-Socket sensor measurement method.

**Figure 8 sensors-25-00832-f008:**
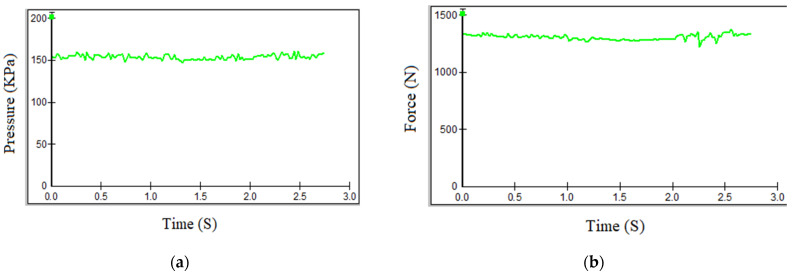
Pressure and force in the posterior region of the exoskeleton using the F-Socket sensor, where (**a**) is pressure and (**b**) is force.

**Figure 9 sensors-25-00832-f009:**
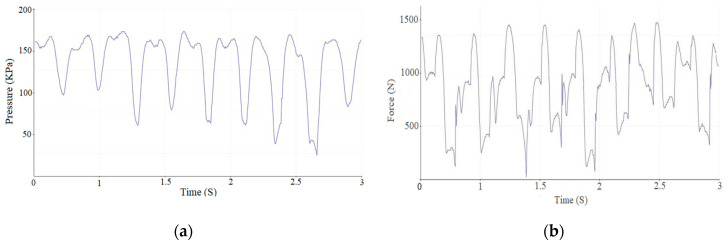
Pressure force in the posterior region of the exoskeleton using EMS sensor, where (**a**) is pressure and (**b**) is force.

**Figure 10 sensors-25-00832-f010:**
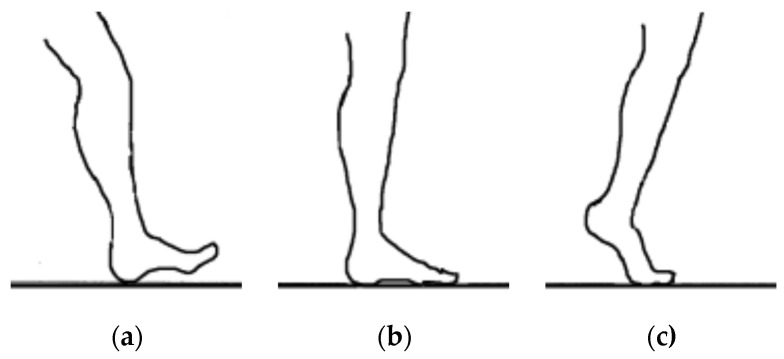
Stance phase events: (**a**) initial contact, (**b**) foot position on the floor, and (**c**) toe-off position.

**Figure 11 sensors-25-00832-f011:**
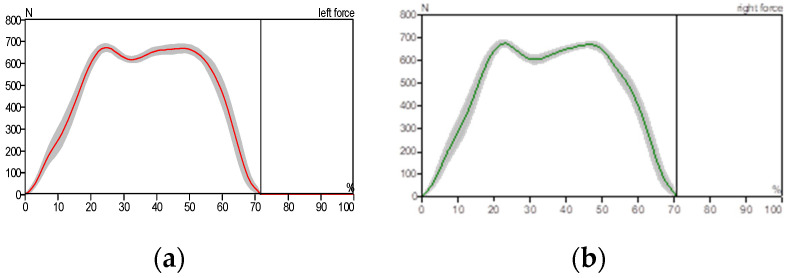
Ground reaction force: (**a**) left foot and (**b**) right foot (healthy individual).

**Figure 12 sensors-25-00832-f012:**
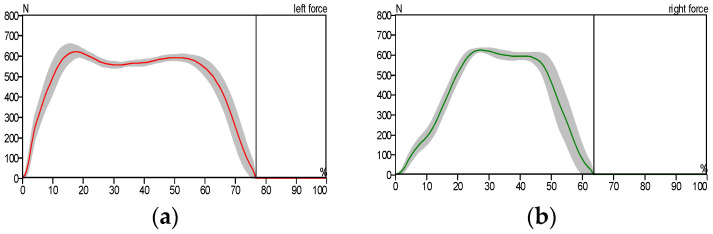
Ground reaction force: (**a**) left foot and (**b**) right foot without wearing the exoskeleton (patient).

**Figure 13 sensors-25-00832-f013:**
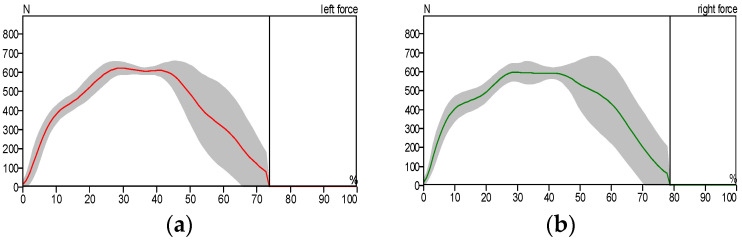
Ground reaction force: (**a**) left foot and (**b**) right foot while wearing the exoskeleton device (patient).

**Figure 14 sensors-25-00832-f014:**
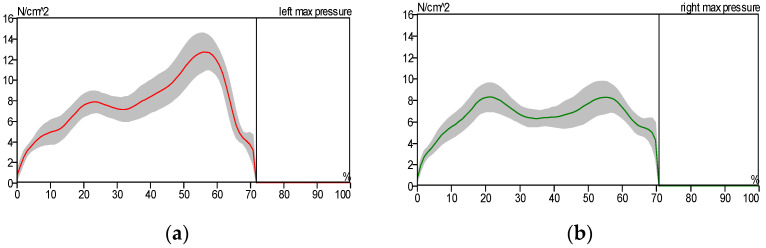
Maximum pressure distribution for healthy subject: (**a**) left foot and (**b**) right foot.

**Figure 15 sensors-25-00832-f015:**
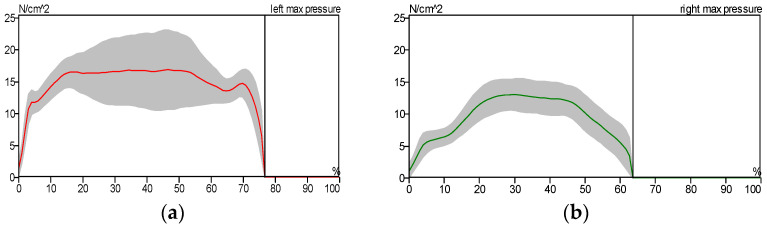
Maximum pressure distribution for the patient without the exoskeleton: (**a**) left foot and (**b**) right foot.

**Figure 16 sensors-25-00832-f016:**
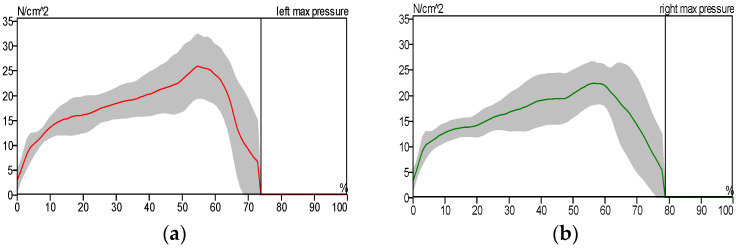
Maximum pressure distribution for the patient wearing the exoskeleton: (**a**) left foot and (**b**) right foot.

**Figure 17 sensors-25-00832-f017:**
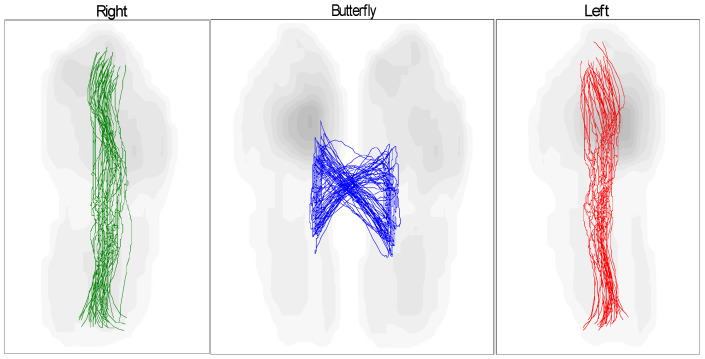
Butterfly parameter of the COP gait cycle for healthy subject.

**Figure 18 sensors-25-00832-f018:**
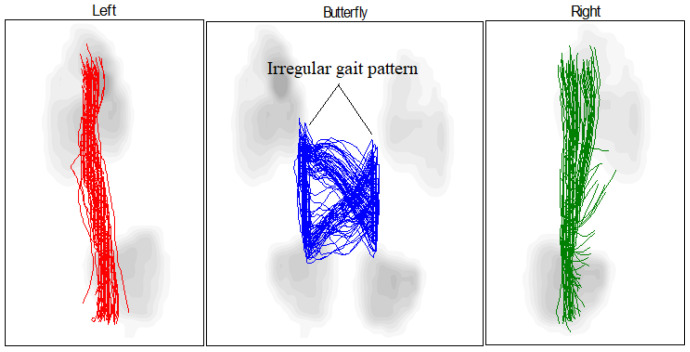
Butterfly parameter of the COP gait cycle for patient not wearing an exoskeleton.

**Figure 19 sensors-25-00832-f019:**
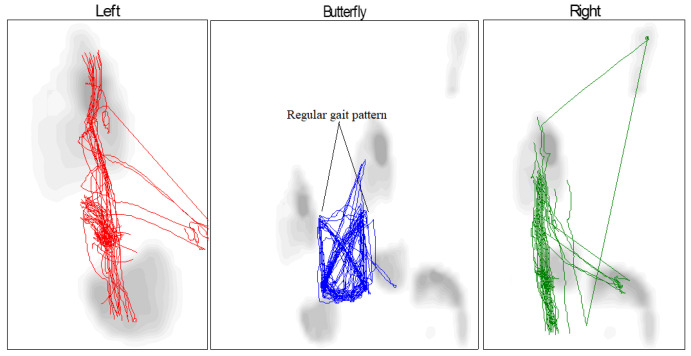
Butterfly parameter of the COP gait cycle for patient wearing an exoskeleton.

**Figure 20 sensors-25-00832-f020:**
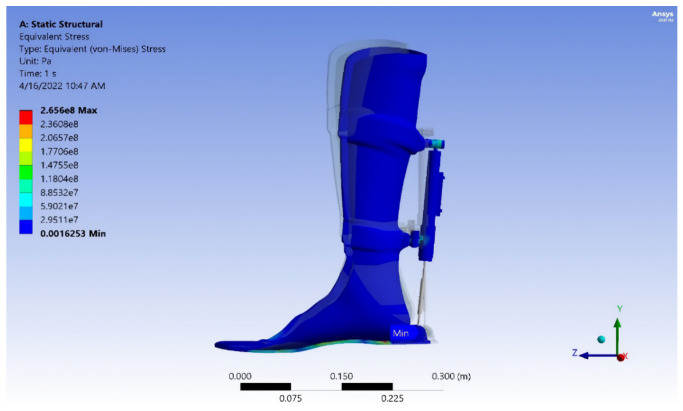
Von Mises stress of the exoskeleton model.

**Figure 21 sensors-25-00832-f021:**
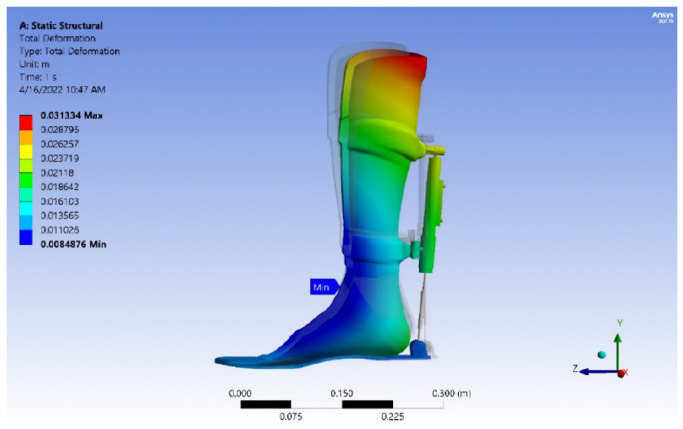
Deformation of the exoskeleton model.

**Figure 22 sensors-25-00832-f022:**
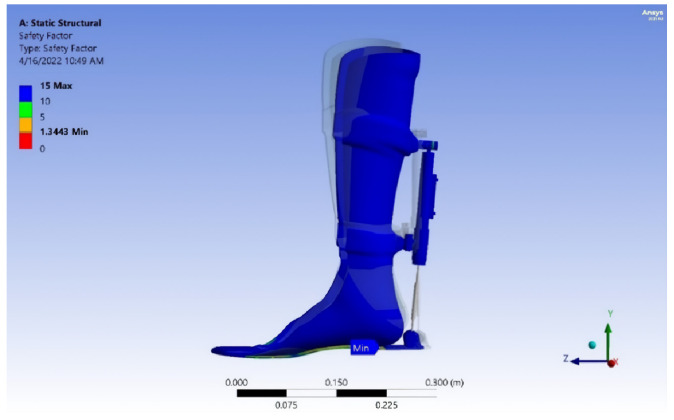
Safety factor of the exoskeleton model.

**Figure 23 sensors-25-00832-f023:**
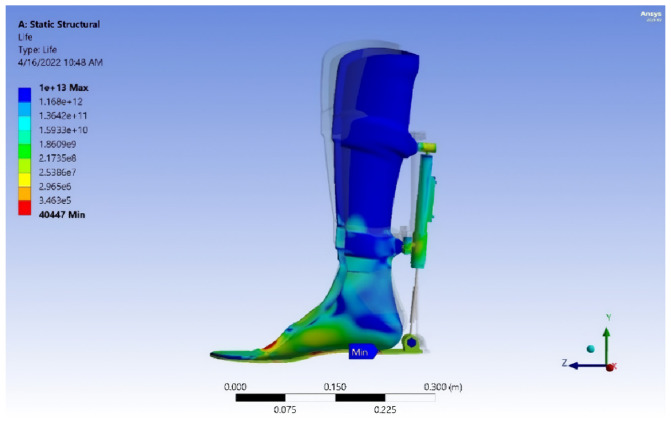
Service life of the exoskeleton model.

**Table 1 sensors-25-00832-t001:** Max-interventional pressure (IP) and force.

Measurement Method	Intervention Pressure, (KPa)	Force (N)
F-Socket sensor	156	1335

**Table 2 sensors-25-00832-t002:** Max-interventional pressure (IP) and force.

Measurement Method	Intervention Pressure, (KPa)	Force (N)
Manufactured EMS sensor	152	1328

**Table 3 sensors-25-00832-t003:** Butterfly parameter for healthy individual.

	Left	Right
Gait line length, mm	238 ± 12	240 ± 15
Single support line, mm	80 ± 14	79 ± 15
Ant/post position, mm	154	
Ant/post variability, mm	9	
Lateral symmetry, mm	−21	
Lateral variability, mm	17	

**Table 4 sensors-25-00832-t004:** Butterfly parameter for patient not wearing an exoskeleton.

	Left	Right
Gait line length, mm	259 ± 11	213 ± 43
Single support line, mm	103 ± 12	103 ± 17
Ant/post position, mm	149	
Ant/post variability, mm	19	
Lateral symmetry, mm	36	
Lateral variability, mm	19	

**Table 5 sensors-25-00832-t005:** Butterfly parameter for patient wearing an exoskeleton.

	Left	Right
Gait line length, mm	113 ± 99	196 ± 110
Single support line, mm	146 ± 31	137 ± 30
Ant/post position, mm	120	
Ant/post variability, mm	35	
Lateral symmetry, mm	−26	
Lateral variability, mm	74	

## Data Availability

The data presented in this study are available in the following published articles: Hussein, T.S.; Izyumov, A.I. Microprocessor System for Measuring Pressure between Orthosis and Foot. Syst. Methods Technol. 2023, 1(57), 80–86. https://doi.org/10.18324/2077-5415-2023-1-80-86 [19]. Hussein, T.S.; Iziumov, A.I. Measurement of Force Parameters for Human Gait Research. Syst. Methods Technol. 2023, 2(58), 30–37. https://doi.org/10.18324/2077-5415-2023-2-30-37 [20].

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
