# Peer review of "Adaptive Exoskeleton Device for Stress Reduction in the Ankle Joint Orthosis"

_sensors, 2025, doi:10.3390/s25030832_

Round 1

Reviewer 1 Report (New Reviewer)

Comments and Suggestions for Authors

This paper presents a exoskeleton for rehabilitaiton of ankle fractures. The exoskeleton is designed and optimized according to the angle and pressure during the ankle motion. Experiments are conducted to validate the exoskeleton. Overall, the paper is not written in a right way to fit the journal standard, and the technical contribution of the paper is too basic. Some comments on the paper are listed as follows:

(1) Although the contribution of the paper have been mentioned in Section 1. Introduction, the novelty of current research is not evident, especially compared with some recent studies on exoskeleton robotics. 

(2) Some explanations should be added in Figs. 1 and 2.

(3) Fig. 3 is not clear. No need to present size in the figure. What do mechanical parts No. 1, 2,... 11 in Fig. 3 (c) mean? 

(4) The variable in Eq. (1) should be italic. This equation is too simple and normaly seen in textbooks, and it should not be emphasized here.

(5) The superscripts and subscripts of the parameters in Eq. (2) should be denoted clearly.

(6) The adaptive load control algorithm is too simple. The authors should declare the quantitative relationship between the exoskeleton parameters and the load pressure at least.

(7) Tables 1 and 2 seem incomplete.

(8) The fonts in Figs. 8 and 9 should be clear and consistent with the manuscript.

(9) What is the shadow in Figs. 11-16.

(10) The conlusions should be rewritten  and consistent with the journal style.

(11) The advantages/limitations/essense of the exoskeleton should be summarized in the experiments.

(12) The reference style does not follow the journal regulation.

Author Response

Reviewer 2 Report (New Reviewer)

Comments and Suggestions for Authors

The authors created a model of an exoskeleton device in SolidWorks, based on data from one healthy and one injured patient. This study aims to enhance rehabilitation by managing load distribution and monitoring joint angles and forces. The technology provides insights into pressure distribution, gait cycles, and foot positioning. An adaptive exoskeleton was developed from data collected by socket sensors, aiding in normalizing gait patterns and reducing ankle joint strain for patients. My recommendation is a major revision to address the following points.

1. How does the exoskeleton device enhance gait parameters in patients with ankle injuries, and which specific metrics can be used to assess and optimize these improvements?

2. What are the main differences in accuracy and performance between a custom-built electromechanical sensor and a commercially available socket sensor?

3. The integration of ultrasensitive pressure sensors (https://doi.org/10.1021/acsami.8b12212) within smart-insole technologies (https://doi.org/10.3390/computation12090184) represents a significant advancement related to the technological findings discussed in this article. Please incorporate these elements into your revised introduction to improve the reach and significance of the findings.

4. How do kinetic properties in exoskeleton devices enhance rehabilitation therapy effectiveness? A discussion on the current state of the art is needed.

5. The authors selected the Butterfly method for its specific advantages. How does the Butterfly parameterization contribute to the development and functionality of the exoskeleton device?

Round 2

Reviewer 1 Report (New Reviewer)

Comments and Suggestions for Authors

The article has been improved well after revision.

Reviewer 2 Report (New Reviewer)

Comments and Suggestions for Authors

The authors have implemented the suggested changes, and the manuscript has also been improved. However, the reference section does not match the revised manuscript. The authors must check/update the references list, format, and correspondence.

Author Response

This manuscript is a resubmission of an earlier submission. The following is a list of the peer review reports and author responses from that submission.

Round 1

Reviewer 1 Report

Comments and Suggestions for Authors

This paper proposes an adaptive exoskeleton device in low-limb rehabilitation. The topic is interesting and important. However, there are some major issues required to be solved.

(1) English writing could be improved and polished. 

For example, Page 2, 'presented'->'present'

'custom'->'customized'

(2) Page 3, 'The aim of the study is the effect of wearing the developed exoskeleton...' could be deleted because the same meaning has been mentioned before this sentence.

(3) Page 3, 'the design most suitable'->'the most suitable design'?

(4) Page 3, 'the reliability of the results is ensured by the use of modern means and methods of research'. Please clarify the detailed information.

(5) Figure 2, all information should be added in English.

(6) Figure 3, the meaning of the numbers should be added.

(7) Page 6, 'k is the stiffness of the spring in N/m'. This definition of k has been mentioned in Page 5. 

(8) Page 6, 'the device is under the maximum force it can withstand (at approximately 1200 N respectively to 120 kg)'. However, in the beginning, it mentioned as 'A normal person walking can exert forces 1.5 times'?

Comments on the Quality of English Language

Moderate editing of English language required.

Reviewer 2 Report

Comments and Suggestions for Authors

Introductory and conclusions sections mention some issues that are not presented in the manuscript. For example, authors mention:

- "Building a structural model of the exoskeleton device control system taking into account the methods of obtaining and processing information related to individual human biopotential"

The control system of the device is not developed in the paper. Authors do not mention the control scheme neither the devices used in control system implementation.

- "Development of the mathematical model of the exoskeleton drive controlled by the block-controller taking into account the individual features of the operator"

The matehematical model of the exoskeleton is not presented. Dynamics of the device is not mentioned along the manuscript.

The manuscript approach is mainly focused in the use of sensors in the exoskeleton. However, development and some obtained measures fron these sensors have been reported previously by the authors. It is not clear the novelty of the presented results in the manuscript.

Minor issues:

- References list must be completly revised to provide all the needed information for each reference and to attend the journal format for different kind of document used as reference.

- Some text in Fig. 2 is in Russian.

- Components shown in Fig. 3 are not described.

Comments on the Quality of English Language

Some statements along the manuscript are difficult to understand. I strongly recommend that the manuscript's grmmar be revised by an english native speaker.

Round 2

Reviewer 1 Report

Comments and Suggestions for Authors

My issues are well solved and it is recommended for publication in Sensors.

Reviewer 2 Report

Comments and Suggestions for Authors

The manuscripts still has some inconsistencies. In specific:

- Authors mention "The novelty of the results presented in this manuscript is demonstrated through the following contributions:

...

Development of a cost-effective and highly accurate force sensor: The study introduces a novel force sensor that is significantly more affordable than existing models, such as the socket-type F sensor, while offering comparable or superior accuracy."

However, authors don't mention any detail about the development of this force sensor. So, this contribution is questionable.

- Authors mention: "The device should simulate the motion of the human ankle joint by moving the EMD (dorsiflexion, plantar flexion) in a range from 25° of plantar flexion to 0° of dorsiflexion in the sagittal plane. This is used to minimise the problem of stiffness in order to speed up the recovery process. "

The proposed range of motion must be justified. According to biomechanics literature, human ankle's range of motion is between 65 and 75°, moving from 10 to 20° of dorsiflexion through to 40–55° of plantarflexion. The useful of the proposed device is questionable with a range of motion of 25°.

- Numbering of the first two equations are wrong (authors number eq. (1) twice). Moreover, units in the first equation are inconsistent (if lenghts are given in mm and force in N, k can not be given in N/m). The second equation to estimate a spring stiffness is wrong, please check any book about design of mechanical elements.

- In lines 224-232 authors mention "To assess structural integrity, finite element method (FEM) software was used to calculate the equivalent fatigue safety stress, and von Mises stress analysis was conducted to evaluate the exoskeleton mode". However, the material property used as strenght parameter is the yield strength which is used in static analysis not in fatigue cases. Moreover, the conditions of the fatigue load are not specified.

- In "Results of kinematic analysis" section, authors report "The maximum ankle angles were 101.93° without the exoskeleton and 98.14° with the exoskeleton" for the foot angles.

It is very difficult to understand how this angle was measured and how it is related with the range of motion of the device (0-25°).

- Some references still present inconsistencies in the format. For example: the use of the word "in" only in some journal articles the use of lower and uppercase in the name of the articles, etc.

Comments on the Quality of English Language

Grammar has been imporved. However, some details must be corrected.
